# Multi-omics analysis reveals metabolic diversity underlying endothelial cell functions

Stephan Durot[1] , Peter F Doubleday[1] , Lydia Schulla[1] , Amélie Sabine[2] , Tatiana V Petrova[2] , Nicola Zamboni[1]

Endothelial cells (ECs) line the vascular system and are key players in vascular homeostasis, yet their metabolic diversity across tissues, vascular beds, and growth states remains poorly understood. This study examines metabolic differences between proliferating and quiescent ECs and compares blood and lymphatic endothelium using proteomics and metabolomics. Our findings indicate that metabolism in quiescent ECs is not dormant but reorganized in a cell-specific manner, with decreased heme intermediates in human umbilical vein ECs and increased branched-chain amino acid catabolism across all quiescent ECs. Consistent with the differences identified in the omics data, perturbation studies revealed that inhibiting enzymes involved in heme, glutamate, fatty acid, and nucleotide biosynthesis led to distinct phenotypic responses in blood and lymphatic ECs. These findings highlight the importance of metabolic pathways in sustaining both proliferating and quiescent ECs and reveal how ECs from different vascular beds rely on distinct metabolic processes to maintain their functional states.

## Introduction

The blood and lymphatic vascular systems are intricate networks that extend throughout the entire body, reaching every organ and tissue (Carmeliet, 2003; Aspelund et al, 2016). These systems are lined by endothelial cells (ECs), which, in a developed vasculature, spend most of their lifespan in a state of quiescence, exiting the cycle of mitotic cell division. However, in response to the need for wound repair or to supply nutrients and oxygen to tissues, ECs become activated and differentiate into migratory tip cells and proliferating stalk cells to form new vessels (Adams & Alitalo, 2007). Following the establishment of new vessels, a return to quiescence is crucial to maintain vascular architecture and homeostasis. Disruption of this elaborate process lead to the progress of various diseases associated to the cardiovascular and lymphatic systems (Adams & Alitalo, 2007; Cao et al, 2012; Morfoisse & Noel, 2019).

Recent studies have highlighted the essential role of metabolism in governing proliferative and quiescent states of ECs, and subsequently form new blood or lymphatic vessels (De Bock et al, 2013a; Eelen et al, 2018; Falkenberg et al, 2019; Dumas et al, 2020; Andrade et al, 2021). Metabolism of all ECs is characterized by a reliance on glycolysis for ATP generation in both proliferating and quiescent states, with a notable decrease in glycolytic flux during quiescence (De Bock et al, 2013b; Wilhelm et al, 2016; Yu et al, 2017; Kalucka et al, 2018). Furthermore, EC metabolism yields key cardiovascular signaling molecules like nitric oxide (NO) and reactive oxygen species (ROS), both promoting migration and proliferation of ECs (Lahdenranta et al, 2009; Bretón-Romero & Lamas, 2014; Morris et al, 2018; Tabrizi et al, 2021; Montenegro-Navarro et al, 2023). In addition, proliferating blood ECs (BECs) upregulate the pentose phosphate pathway (PPP), tricarboxylic acid (TCA) cycle, serine and glutamine metabolism and the electron transport chain (ETC) for nucleotide biosynthesis and biomass formation (Falkenberg et al, 2019). Quiescent BECs, on the other hand, prioritize the PPP, TCA cycle, and fatty acid $\beta$-oxidation (FAO) for redox homeostasis (Falkenberg et al, 2019). Recent studies also showed an increase in TCA cycle anaplerosis through upregulated glutamine metabolism and decrease of glycolysis in BECs that encounter wall shear stress in vivo (Simões-Faria et al, 2025). Metabolism of lymphatic ECs (LECs) is less well studied; however, FAO and ketone body oxidation are known to play crucial roles in LEC differentiation and proliferation. Upregulation of FAO by lymphatic transcription factor PROX1 leads to epigenetic-based increased expression of lymphatic genes while upregulation of ketone body oxidation induced a proliferative state in LECs (Wong et al, 2017; García-Caballero et al, 2019; Simeroth & Yu, 2024).

Whereas the role of metabolism in regulating the growth states and identities of ECs has become clearer, previous studies have some limitations. Much research on EC metabolism has focused on specific metabolic pathways or has used human umbilical vein ECs (HUVECs) as a universal model, which may not fully represent the complexity of endothelial biology in various contexts. Moreover, transcriptomic studies on organ-specific murine ECs revealed distinct mRNA expression profiles of metabolic proteins, further

[1]Institute of Molecular Systems Biology, ETH Zürich, Zürich, Switzerland    [2]Department of Oncology, University of Lausanne and Ludwig Institute for Cancer Research, Epalinges, Switzerland

Correspondence: zamboni@imsb.biol.ethz.ch

highlighting the metabolic diversity among ECs (Kalucka et al, 2020; Paik et al, 2020).

Our study investigates the metabolic characteristics of four different types of ECs—two from the cardiovascular system and two from the lymphatic system—in both proliferating and quiescent states. We used untargeted semi-quantitative proteomics and metabolomics to build upon and enhance previous transcript-level research, gaining a deeper understanding of the metabolic properties of all the conditions we studied. Our findings highlight notable differences between growth states, vascular beds, and cell types, demonstrating the detailed nature of ECs' metabolism.

# Results

### Distinct proteomic signatures define endothelial cell line identities and states

We began by studying the growth and quiescence induction in four different types of endothelial cells from the two different vascular beds and three different tissues: HUVECs, human dermal blood endothelial cells (HDBECs), human dermal lymphatic endothelial cells (HDLECs), and human intestinal lymphatic endothelial cells (iLECs). Quiescence was induced through contact inhibition and mitogen reduction. We determined the percentage of proliferating cells in each cell line by measuring EdU incorporation into newly synthesized DNA 24 h after EdU addition, which serves as an indicator of mitotic cell division (Fig S1A) (Salic & Mitchison, 2008). Cells that did not incorporate EdU after 24 h were labelled as quiescent, in a G0-arrested state. Two days after seeding, each cell line showed a fraction of EdU-negative (quiescent) cells, with HUVECs having the lowest percentage at 5% and HDBECs having the highest at 20%. After 5 d in continuous culture, the percentage of quiescent cells increased to over 80% in HUVECs, HDLECs, and iLECs, and more than 70% in HDBECs (Fig 1A). By measuring the viability and performing reseeding experiments with each cell type, we show that more than 85% of cells are viable throughout the whole experiment and enter normal cell cycle after reseeding, proving that cells are indeed in a functional quiescent state (Fig S1B and C).

To understand the molecular mechanisms that determine the states and identities of ECs, we conducted label-free quantitative proteomics analyses on all four cell lines from day 2 to day 7 post-seeding. We identified 8,098 protein groups with a 1% protein group-level false discovery rate (FDR) cutoff (Table S1). To explore the proteomic patterns that distinguish cellular identity and the transition to quiescence in different types of ECs, we focused on day 2 and day 5 after seeding, as these time points consistently represent active proliferation and quiescence states across various cell lines. A comparison between samples from day 5 and day 2 revealed a down-regulated, cell cycle-related core proteomic signature in quiescent ECs. Proteins involved in DNA replication, translation, and the cell cycle, such as CDK1, MCM2, MCM3, MCM7, and LARP4, were down-regulated in quiescent cells (Fig S2A) (Madine et al, 1995; Tsao et al, 2004; Santamaría et al, 2007; Küspert

et al, 2015; Petryk et al, 2018). In addition, we observed significant changes in the expression of proteins related to vessel formation, maintenance, and cellular quiescence. Proteins associated with extracellular matrix organization and adhesion (CCN1, NID1), platelet activation and coagulation (VWF, MMRN1, TFPI), autophagy (ACP2, LIPA, GAA, TPP1), and the suppression of inflammation and senescence (NTN4) were all up-regulated more than twofold during quiescence (Fig S2A) (Nightingale & Cutler, 2013; Maroney & Mast, 2015; Grijalva et al, 2016; Kim et al, 2018; Smith et al, 2019; Leatherdale et al, 2021; Li et al, 2021; Zhang et al, 2021; Cheng et al, 2022; Fu et al, 2022).

Beyond the foundational differences in protein expression, our use of principal component analysis and Spearman's correlation further identified different proteomic patterns dependent on cell types and their proliferative states (Figs 1B and S2B). Whereas proliferating ECs display diverse proteomic patterns, quiescence ECs from the same vascular beds share similar proteomic profiles. This observation led us to hypothesize that the proteomic distinctions may reflect the unique physiological functions of quiescent lymphatic and blood ECs in maintaining tissue homeostasis. To identify the pathways pivotal to these functions, we conducted a pathway enrichment analysis with proteins changing significantly between day 5 and day 2. The analysis revealed higher abundances of proteins in quiescence in 105 pathways and lower abundances of proteins in quiescence in 186 pathways in at least one cell line (FDR < 0.05, Table S2). Proteins with lower abundances in quiescence prominently include those involved in the cell cycle and nucleotide metabolism, but also extended to rRNA processing, translation, and the metabolism of amino acids and their derivatives, encompassing all pathways responsible for the biosynthesis and degradation of amino acids (Table S2).

During quiescence, proteins from different pathways show varying levels of increased abundance in specific types of endothelial cells. For example, in HDBECs, the most enriched pathways are related to membrane trafficking, vesicle-mediated transport, and asparagine N-linked glycosylation (Table S2). On the other hand, the top five enriched pathways in HDLECs relate to the TCA cycle, ETC, and Complex I biogenesis (Table S2). Notably, proteins in metabolic pathways are enriched in all four cell lines during quiescence, emphasizing the crucial role of metabolism in both inducing and maintaining quiescent states. To further examine which aspects of the metabolic network are essential for quiescence, we focused on the 36 metabolic pathways that were enriched in at least one cell line (FDR < 0.05) (Fig 1C). Among these, proteins in eight pathways show consistently lower abundance, including nucleotide biosynthesis and the metabolism of amino acids and derivatives. Higher abundant proteins are observed in pathways related to mitochondrial FAO, aligning with the previously reported increase of FAO transcript levels in quiescent HUVECs for vasculoprotection through redox homeostasis (Kalucka et al, 2018). Moreover, all quiescent ECs show an increase in branched-chain amino acid (BCAA) catabolism, which suggests an important role similar to that described in hematopoietic stem cells through a mechanism that involves the PPM1K phosphatase or regulation of OGDH activity, as described previously in HUVECs (Liu et al, 2018; Andrade et al, 2021).

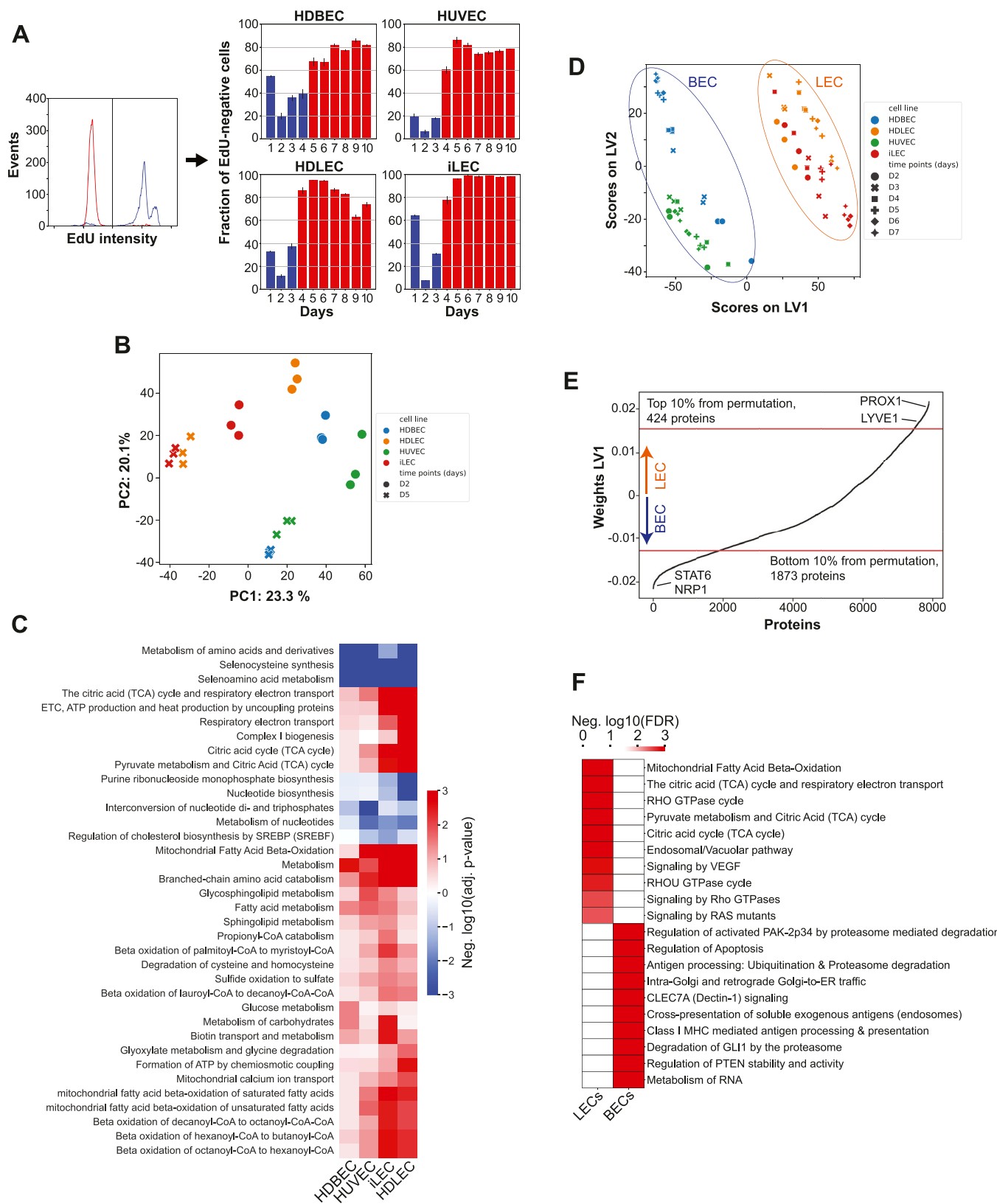

**Figure 1. Endothelial cell lines identities and states are defined by distinct proteomic signatures.**
**(A)** Fraction of EdU-positive cells determined by FACS. **(B)** Principal component analysis of day 2 (proliferating) and day 5 (quiescent) proteomics samples. **(C)** Pathway enrichment analysis of proteomics data of quiescence (day 5) versus proliferation (day 2) samples, hierarchically clustered. Negative values correspond to decreased abundances in quiescence, positive values to increased abundances in quiescence. **(D)** Partial least squares discriminant analysis (PLS-DA) of day 2 to day 7 proteomics

Next, we explored the proteomic patterns that determine the identities of ECs. To that end, we used partial least squares discriminant analysis (PLS-DA) to identify proteins whose expression levels distinguish between lymphatic (iLECs and HDLECs) and blood endothelial cells (HDBECs and HUVECs) (Fig 1D and Table S2). This analysis allowed us to examine general, state-independent proteomic patterns underpinning LEC and BEC identities, including grouping individual cell types across all states. We used a permutation test on the first component weights of the PLS-DA, which differentiated BECs from LECs, to identify proteins associated with LECs and BECs. Using a 10% cutoff, we found that the list comprises 424 proteins for LECs and 1,873 proteins for BECs. These lists include well-known markers such as PROX1, LYVE1, FLT4, NRP2, and ITGA9 for LECs, and STAT6, NRP1, MMP14, ICAM2, CD93, and ECE1 for BECs, suggesting that the in vitro LEC and BEC cultures in this study still express differentiation programs that are crucial for vessel formation and homeostasis in vivo (Figs 1E and S2C and D, Table S2) (Podgrabinska et al, 2002; Wilting et al, 2002; Huang et al, 2005; Tammela & Alitalo, 2010; Galvagni et al, 2016; Gauvrit et al, 2018; Wong et al, 2018; Han et al, 2019; Gao et al, 2022). To determine the processes that are characteristic of each vascular bed, we conducted a pathway enrichment analysis with these two lists. We found that BEC-associated proteins are enriched in 153 pathways, whereas LEC-associated proteins are enriched in 78 pathways (FDR < 0.05) (Table S2). Notably, metabolic pathways are among the top enriched pathways in LECs but not in BECs. For example, fatty acid $\beta$-oxidation, TCA cycle, and the respiratory ETC are the two most enriched pathways in LECs, which underscore their vital roles in lymphangiogenesis and LEC specification and maintenance (Fig 1F) (García-Caballero et al, 2019; Ma et al, 2021). In addition, we found that whole metabolism (FDR = 0.004), glyoxylate metabolism, and glycine degradation (FDR = 0.005), as well as catabolism of lysine (FDR = 0.02) and branched-chain amino acids (BCAAs) (FDR = 0.02) are significantly enriched in LECs but not in BECs across all growth states (Table S2). Taken together, proteomic analysis of ECs revealed that certain metabolic programs, such as decreased nucleotide metabolism and increased FAO and BCAA catabolism, are conserved and distinguish quiescence from proliferation across ECs. Furthermore, proteomic patterns specific to individual EC types diverge significantly, reflecting the unique physiological functions and identities of EC types.

## Functional analysis by metabolomics

To acquire direct evidence of metabolic differences across EC types and states, we measured extracellular metabolites' uptake and secretion rates during the transition from proliferation to quiescence. We took daily measurements by exchanging the growth medium every 24 h and analyzing the cellular supernatants' content through mass spectrometry before the next medium replacement (Fig S3A, Table S3) (Fuhrer et al, 2011). In total, we identified 521 putative metabolites and determined their respective uptake and secretion rates (Fig S3B and C). Because of these rates depend on each compound's initial abundance in the medium, we further calculated the z-score across all conditions to highlight patterns characteristic of EC types or states (Fig 2A). When comparing proliferative and quiescent cells, our focus was on day 2 and day 5, respectively.

One interesting observation is that ECs remain metabolically active even when their growth is arrested. Quiescent ECs secrete lactate, uracil and (iso)citrate and take up pyruvate, ascorbate, fumarate, and adenine from the growth medium (Fig 2A). Except for uracil, these metabolites are also secreted or taken up by all cell types in proliferation. However, uptake and secretion rates of other metabolites, like hexose and amino acids, are more diverse amongst the different cell types and states. For example, HUVECs have high uptake rates of hexose, serine, cysteine, methionine, tyrosine, and arginine in proliferation and quiescence. On the other hand, HDBECs secrete many amino acids in both growth states and only take up hexose in quiescence. HDLECs take up hexose in proliferation and quiescence, whereas iLECs only take up hexose in proliferation. Both, HDLECs and iLECs consume several amino acids during proliferation and secrete some amino acids (alanine, proline, and aspartate) in quiescence (Fig 2A). Overall, extracellular metabolomics data suggest that there is not a general decrease in metabolic activity in quiescent ECs, but rather a reorganization of metabolism to meet specific cellular demands.

Next, we performed untargeted metabolomics of the intracellular space to investigate intracellular metabolic dynamics, identifying 1,413 putative metabolites across all cell lines (Table S4). Principal component analysis and Spearman's correlation analysis revealed distinct intracellular metabolic patterns among cell types and states (Figs 2B and S4A). In proliferation, metabolic patterns appear more cell-specific, whereas the differences in quiescence are more subtle. Overall, 63 metabolites were found to be significantly changed in at least one cell type between quiescence and proliferation (Fig S4B). Most of these metabolites have larger pool sizes in quiescence; whether this results from accumulation or activation cannot be determined from this data and needs experimental validation. To interpret these changes in a network context, we conducted a pathway enrichment analysis with all metabolites that were deemed significant to examine more general metabolic changes (Fig S4C). A large portion of pathways have increased abundances of intermediates in quiescence, following the trend observed on the metabolite level. For example, intermediates of TCA cycle, glycolysis, phospholipid biosynthesis, aspartate metabolism, or pterin biosynthesis are consistently increased in quiescence across all cell types. Yet, we also observed cell type-specific lower abundances of metabolites in quiescence in several pathways, such as glycine and serine metabolism, glutamate metabolism, pyruvate metabolism or lysine degradation in HUVECs and iLECs or arginine and proline metabolism, porphyrin metabolism, glutathione metabolism, pyrimidine metabolism, or urea cycle in HUVECs.

---

samples showing the discrimination between LECs and BECs among component 1. **(E)** Weights of the first component of the PLS-DA. Each protein has a weight that corresponds to its cell type-related information. A bootstrapping approach was applied to determine the 10th and 90th percentile, associated with BEC and LEC identity, respectively. **(F)** Pathway enrichment analysis of LEC- and BEC-associated proteins, showing the top 10 enriched pathways for both cell types.

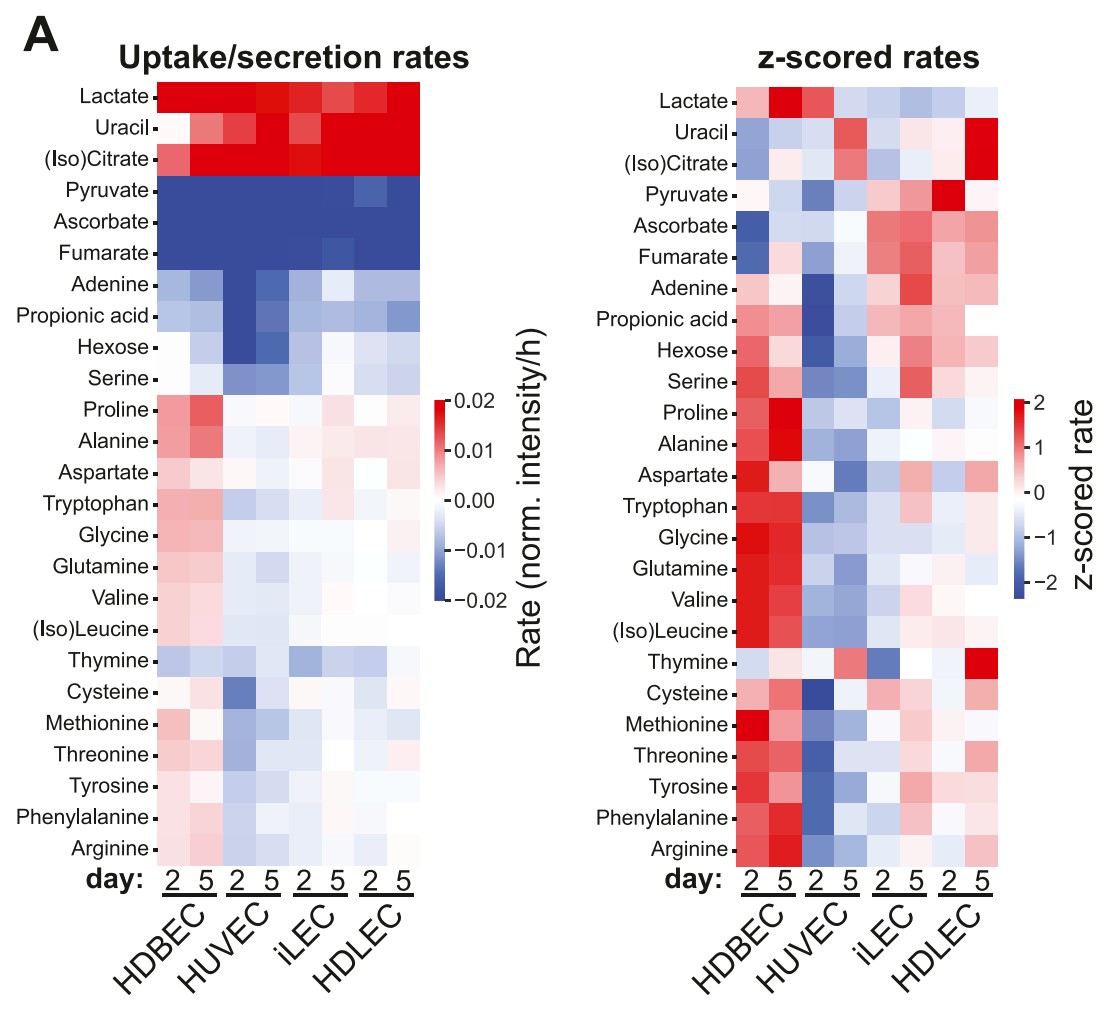

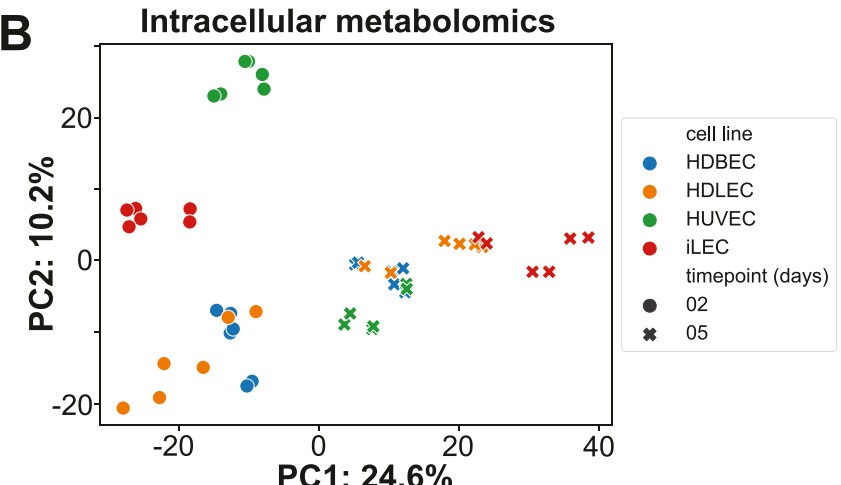

**Figure 2. Distinct metabolic programs underlie quiescence induction and maintenance processes across different cell lines.**
**(A)** Left: normalized ion intensity changes per hour for detected amino acids and growth media components. Negative rate corresponds to uptake, positive rates to secretion of metabolites. Right: z-scored uptake/secretion rates. z-scoring was performed over metabolites to illustrate relative changes between cell lines.
**(B)** Principal component analysis of intracellular metabolites of each cell line in proliferation (D2) and quiescence (D5).

In summary, quiescent ECs, despite exiting the proliferative cycle, remain metabolically active but undergo significant metabolic reorganization to meet their specific physiological demands. Instead of a general reduction in metabolic activity, quiescent ECs show increased expression of proteins involved in FAO and BCAA catabolism, along with higher abundances of intermediates in the TCA cycle, glycolysis, aspartate, pterin, and other biosynthetic pathways. In addition, we observed cell type-specific enhanced regulation of porphyrin, glycine and serine, glutathione and glutamate metabolism in quiescence. This adaptive metabolic shift likely supports the needs of quiescent ECs in maintaining vessel integrity and function.

### Cell types and states are associated with distinct metabolic dependencies

From our multi-omics analysis, we hypothesized that distinct endothelial cell types may rely on different metabolic pathways to support quiescence and proliferation. To verify this hypothesis in a functional context, we pharmacologically inhibited pathways of interest in iLECs and HUVECs and analyzed the impact on proliferation, migration, and sprouting (Fig S5, Table S5). We chose drugs targeting the TCA cycle and ETC because these pathways show higher protein and metabolite abundance in quiescent HUVECs and iLECs. In addition, these two pathways are linked to LEC identity in our proteomics screen, suggesting a potential higher susceptibility of iLECs compared with HUVECs. We chose multiple drugs that target pathways which are similarly regulated in quiescent HUVECs and iLECs, to assess whether the two cell types have the same susceptibility to perturbations in these pathways. These pathways include glutamate and glycine and serine metabolism, which are targeted due to reduced intermediates in quiescent HUVECs and iLECs, as well as pterin biosynthesis and lipid metabolism, which are more abundant in quiescent cells. Additional targets include purine and pyrimidine biosynthesis, as these are down-regulated at the protein level in quiescent iLECs and HUVECs. Conversely, quiescent HUVECs, but not iLECs, display increased purine and decreased pyrimidine intermediates. The urea cycle, glutathione metabolism, and porphyrin/heme metabolism are also targeted, as their intermediates are lower only in quiescent HUVECs. Lastly, the pentose phosphate pathway is targeted due to its general vital role in redox homeostasis. This results in a total of 14 inhibitory drugs that were used for validation experiments (Table S5).

We started by assessing the effect of the 14 drugs on proliferation and migration. Migration is a normal function of healthy ECs, making migration assays useful for studying key processes involved in (lymph)angiogenesis (Liang et al, 2007). By perturbing quiescent ECs and conducting migration assays, metabolic vulnerabilities in quiescence and the transition to migration can be identified. Migration assays involve applying a scratch to a 2D cell culture and measuring the time it takes for the cells to close the wound. We conducted the migration assays in two different ways: cells were either exposed to the drugs 5 d before and after applying a scratch to study long-term, quiescence-dependent effects, or only after the scratch was made for short-term, quiescence-independent effects (Fig S5).

Overall, we observed diverse growth and migratory phenotypes in proliferating and migrating HUVECs and iLECs upon treatment with similar concentrations of the 14 drugs for both cell types (Fig 3A). Drugs that target nucleotide biosynthesis, folate metabolism, fatty acid synthesis, and the transport of pyruvate into mitochondria all decrease proliferation and/or migration of HUVECs and iLECs. iLECs have generally lower migration rates when the cells are treated for 5 d before the scratch assay, indicating that quiescent iLECs are more sensitive to perturbations in these pathways. Furthermore, blockage of the oxidative deamination of glutamate to α-ketoglutarate by R162 and pathways related to ROS and NO metabolism (targets of succinylacetone, rotenone, dimethyl fumarate, and buthionine sulfoximine) exhibit divergent dependencies of HUVECs and iLECs on these pathways for establishing and maintaining certain phenotypes.

### Inhibition of nucleotide, folate and lipid metabolism disrupts proliferation and migration of HUVECs and iLECs

Our proteomics data showed that fatty acid synthase (FASN) expression is significantly increased in proliferating HUVECs and iLECs relative to quiescent cells (Fig 3B). FASN has previously been reported to be critical for proliferation and migration of HUVECs and LECs (Wei et al, 2011; Bastos et al, 2017; Bruning et al, 2018). We therefore wondered whether we could reproduce the previously described defects in proliferation and migration of HUVECs and iLECs with our experimental setup and whether HUVECs and iLECs exhibit similar sensitivities to FASN inhibition within our setup. Indeed, FASN inhibitor C75 (Rae et al, 2015) stopped proliferation of HUVECs and iLECs, and significantly reduced migration in HUVECs and iLECs after short- and long-term treatment, with iLECs being more sensitive to long-term and HUVECs to short-term treatment (Fig 3C and D). These results show that we can reproduce known phenotypes with our experimental setup and that FASN inhibition decreases proliferation and migration of HUVECs and of iLECs, potentially through mTOR malonylation and inactivation in proliferation and reduced eNOS palmitoylation and activity in migration (Wei et al, 2011; Bruning et al, 2018).

In our proteomics screen, we found that all quiescent ECs have increased expression of TCA cycle enzymes (Fig 1D). To confirm the significance of this observation, we tested the impact of mitochondrial pyruvate uptake with UK5099. The treatment reduced growth and migration in HUVECs, and to some extent in iLECs, although the effect was only observed after long-term treatment but with no statistical significance (*P* value = 0.08) (Fig 3C and D). A recent study showed that upregulation of the mitochondrial pyruvate carrier (MPC) in adult neural stem cells is necessary to maintain quiescence by sustaining the TCA cycle and OXPHOS (Petrelli et al, 2023). Based on our observations, this mechanism is potentially also involved in endothelial quiescence behind their previously described role in redox homeostasis.

Pathway enrichment analysis on the protein level showed that proteins in nucleotide biosynthesis are lower expressed in quiescent HUVECs and iLECs (Fig 3B), but surprisingly the metabolite levels in quiescence differ between HUVECs and iLECs (Fig S4C). We examined the effects of 5-Fluorouracil (5-FU), Pemetrexed and Methotrexate on proliferation and migration of HUVECs and iLECs

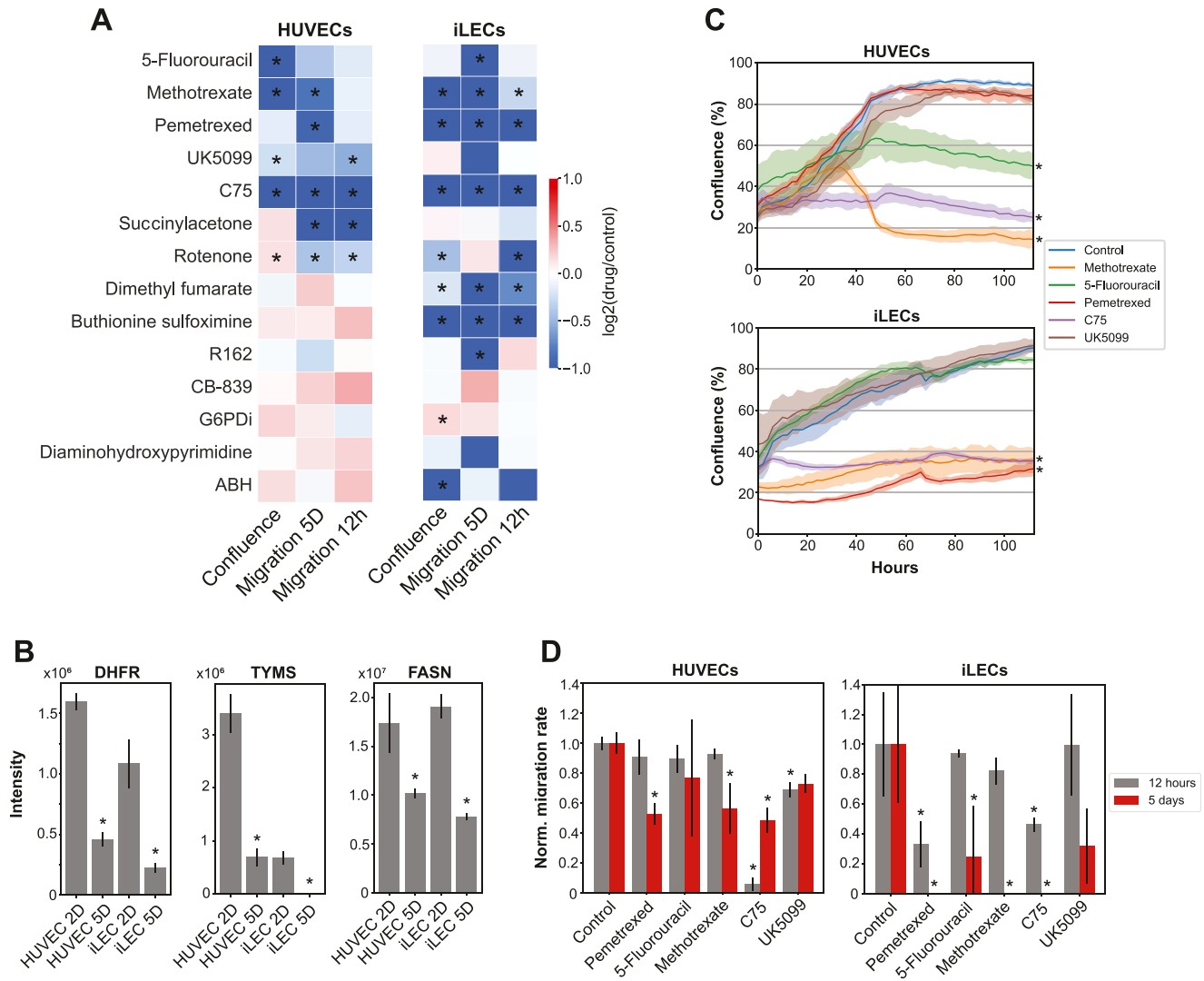

**Figure 3. HUVEC- and iLEC-specific metabolic programs are susceptibilities during proliferation and quiescence induction.**
**(A)** Overview of the phenotypic consequences after treatment with the 14 drugs. Migration 5D refers to the migration rate after 5 d of drug treatment and migration 12 h to the migration rate without prior drug treatment. n = 3 replicates for each perturbation and cell line in all measurements. *P*-values were determined using a Welch's *t* test. *$P$-value < 0.05. **(B)** Expression levels of TYMS, DHFR and FASN. *$Q$-value < 0.05 comparing 5D with 2D abundances. Data are represented as mean ± SD. **(C)** Growth curves of HUVECs and iLECs treated with drugs targeting nucleotide biosynthesis, folate metabolism, lipid biosynthesis, and the mitochondrial pyruvate transport. n = 3 replicates. *P*-values were determined using a Welch's *t* test. *$P$-value < 0.05. Data are represented as mean ± SD. **(D)** Migration rates of HUVECs and iLECs, normalized to the respective control. n = 3 replicates. *P*-values were determined using a Welch's *t* test. *$P$-value < 0.05. Data are represented as mean ± SD.

to assess whether the two cell types respond similarly to inhibition of nucleotide biosynthesis and folate metabolism. 5-FU and Pemetrexed target thymidylate synthase (TYMS), whereas Methotrexate and Pemetrexed target dihydrofolate reductase (DHFR) (Hirata et al, 1989; Longley et al, 2003; Adjei, 2004). Our findings revealed that Methotrexate and 5-FU inhibit growth of HUVECs and Methotrexate and Pemetrexed growth of iLECs (Fig 3C). Moreover, after long-term treatment, Pemetrexed and Methotrexate significantly reduced migration in iLECs, and to a lesser extent, in HUVECs (Fig 3D). 5-FU also reduced migration in iLECs, whereas HUVECs exhibited a nonsignificant decrease. Short-term Pemetrexed treatment significantly reduced migration in iLECs, but not in HUVECs. These results indicate that a functioning nucleotide and

folate metabolism is vital for proliferation as well as quiescence because of long-term treatment of drugs has a much stronger detrimental effect on migration, especially in iLECs.

We hypothesized that the differences in resistance could be explained by the varying expression of the target enzymes TYMS and DHFR. For instance, the expression levels of TYMS are ~6 times higher in proliferating HUVECs than in iLECs, possibly indicating a greater reliance of proliferating HUVECs on TYMS and heightened vulnerability to inhibition with 5-FU (Fig 3B). However, despite DHFR expression being 30% lower in proliferating iLECs compared with HUVECs, iLECs show a similar vulnerability to Methotrexate treatment as HUVECs (Fig 3B). This suggests that susceptibility to drugs does not necessarily align with expression levels of the

target enzymes, but also depends on other mechanisms, such as differences in metabolic fluxes, compensatory pathways or drug metabolism and efflux.

## Disrupted glutamate levels impair migration and sprouting in an EC-specific manner

In our metabolomics analysis, we observed a strong decrease of glutamate intermediates and uptake of glutamine in quiescent HUVECs but not iLECs. BECs, but not LECs, are known to depend on glutamine and glutamate metabolism for proliferation and migration, mainly through anaplerosis and as nitrogen source for transamination reactions (Huang et al, 2017; Kim et al, 2017). Conversely, a recent study showed that BECs also rely on glutamine-fueled anaplerosis under pro-quiescent wall shear stress situations in vivo (Simões-Faria et al, 2025). To assess the functional relevance of the observed metabolic differences in HUVECs and iLECs, we tested their sensitivity to CB-839, a glutaminase (GLS1) inhibitor, and R162, a glutamate dehydrogenase (GLUD1) inhibitor. Due to higher glutamine uptake and stronger reduction of glutamate metabolism intermediates in quiescent HUVECs, we expected higher susceptibility of HUVECs to these drugs in migration. CB-839 and R162 do not affect the proliferation of HUVECs and iLECs (Fig 4A). Migration of HUVECs and iLECs is not impaired by CB-839 (Fig 4B), but a striking 60% decrease in migration of iLECs was observed for long-term R162-treated iLECs, whereas HUVECs exhibit a nonsignificant 16% decrease (Fig 4B).

To verify whether the action of R162 is caused by reduced production of $\alpha$-ketoglutarate by GLUD1, we attempted to rescue the phenotype with addition of the cell-permeable dimethyl-$\alpha$-ketoglutarate (dmKG). In line with the R162 effect, the rescue is substantial in iLECs and marginal in HUVECs (Fig 4C). In both cases, however, the reproducibility was not sufficient to attain statistical significance (Fig 4C). Strikingly, dmKG alone significantly impairs iLEC migration and, to a much lesser, nonsignificant extent, HUVEC migration (Fig 4C). To assess if the different migratory responses also affect angiogenesis, we conducted a sprouting assay with spheroids. We measured the total sprout length after exposing them to CB-839, R162, dmKG, or R162 in combination with dmKG for 3 d and determined the total sprout length (Tetzlaff & Fischer, 2018) (Fig 4D). R162 reduces sprouting by 30% in HUVECs and iLECs, and supplementation of dmKG aggravated the defect to 70%. Treatment with CB-839 increases the sprouting similarly in both ECs. Together, iLECs differ from HUVECs by having a higher sensitivity to excess glutamate and $\alpha$-ketoglutarate levels, which consequently impairs migration. Moreover, the sprouting results indicate a role of glutamate metabolism on angiogenesis beyond its isolated effect on EC migration.

## EC-specific dependencies on nitric oxide and ROS metabolic pathways for migration and sprouting

Five drugs in our screen directly or indirectly interfere with ROS and nitric oxide (NO) metabolism as part of their mechanism of action (Sassa & Kappas, 1983; Li et al, 2003; Nishizawa et al, 2018; Ghergurovich et al, 2020; Ocaña et al, 2023) (Fig 5A). These drugs were chosen due to decreased levels of serine intermediates in quiescent ECs, HUVEC-specific decreased abundance of glutathione and porphyrin/heme intermediates and increased abundance of proteins and intermediates of the ETC in quiescent HUVECs and iLECs.

We found that the growth of iLECs, but not HUVECs, is reduced by rotenone, buthionine sulfoximine (BS) and dimethyl fumarate (DF) (Fig 5B). The former blocks Complex I of the mitochondrial ETC, leading to elevated ROS levels (Li et al, 2003). Buthionine sulfoximine inhibits de novo glutathione biosynthesis and dimethyl fumarate de novo serine biosynthesis which, in turn, fuels the biosynthesis of heme, a cofactor of ROS-clearing catalases and eNOS (Nishizawa et al, 2018; Vandekeere et al, 2018; Ocaña et al, 2023). In contrast, HUVECs are affected in migration rate by rotenone and succinylacetone, which inhibits aminolevulinic acid dehydratase (ALAD) in heme/porphyrin biosynthesis (Fig 5C) (Sassa & Kappas, 1983). In iLECs, migration is affected only when rotenone is added right after the scratch assay, or upon treatment with DF and BS, which is likely to be a consequence of reduced proliferation. Moreover, we observed that the sensitivity of HUVECs and iLECs to the five drugs does not correlate with the expression of their primary targets. For example, expression of ALAD, the target of SA, and GCLC, the target of BS, is higher in quiescent HUVECs, but only SA leads to decreased migration (Fig S6). Furthermore, iLECs have an increased expression of Complex I proteins (NDUFS1, NDUFV2), but migration is only reduced in HUVECs and not iLECs upon treatment with rotenone for 5 d (Fig S6).

The role of NO and ROS in promoting angiogenesis is well known. To verify if the different migratory responses also affect (lymph-)angiogenesis, we conducted a sprouting assay with spheroids. The experiments were run simultaneously with the experiments in the previous chapter. We measured the total sprout length after exposing them to rotenone and succinylacetone for 3 d (Fig 5D). In HUVECs, both drugs reduced the sprout length by 75% and 85%, respectively. In iLECs, only rotenone had a noticeable effect. Overall, proliferating HUVECs are more resistant to changes in redox and NO pathways compared with iLECs. However, they depend on balanced ROS levels and heme availability for migration and sprouting.

We wondered whether balanced ROS levels or heme availability are needed for HUVECs and iLECs migration and sprouting. Thus, we supplemented SA- and Rotenone-treated HUVECs and iLECs with either diethylamine NONOate, a NO donor, or dimethylthiourea (DMTU), a hydroxyl radical scavenger. NONOate and DMTU rescue SA-induced migratory defects in HUVECs and promote migration of SA-treated iLECs in the short- and long-term (Fig 5E and F). NONOate does not rescue Rotenone-induced migratory effects in HUVECs but boosts migration of short-term Rotenone-treated iLECs. DMTU slightly, but nonsignificantly increases migration rate of HUVECs treated with Rotenone for 12 h after the scratch was made (Fig 5F). In addition, when DMTU and Rotenone are added to iLECs after the scratch is made, we detect a twofold increase in migration rate and a nonsignificant 40% increase when DMTU is added to iLECs treated for 5 d with Rotenone (Fig 5F). NONOate alone significantly increases migration rate of both ECs when treated for 5 d before the scratch assay, but not when added right after the scratch is made (Fig 5G), potentially through activation of cGMP-Rho GTPase signaling that exerts its effect on

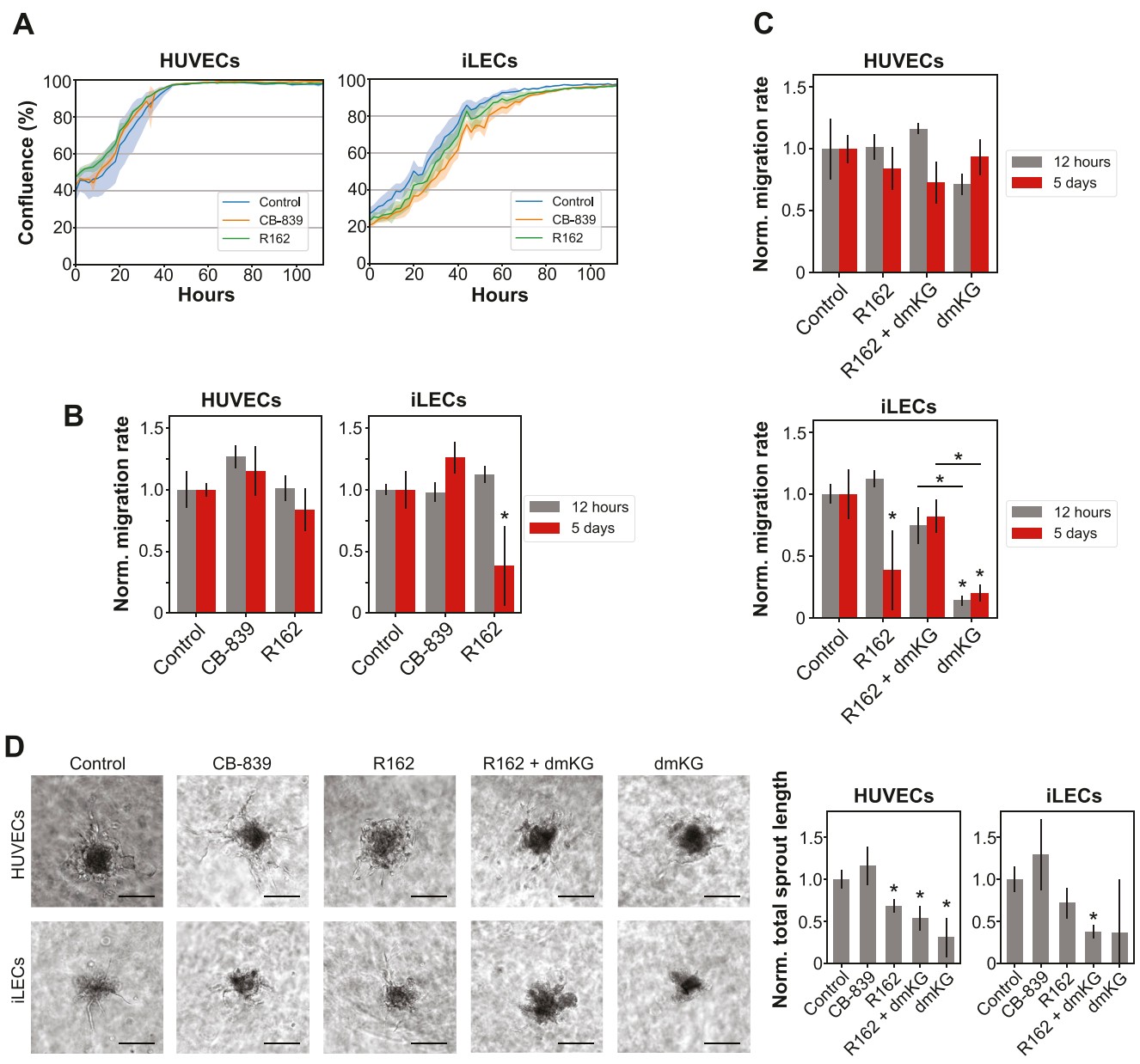

**Figure 4. Disrupted glutamate levels impair migration and sprouting in an EC-specific manner.**
**(A)** Growth curves of HUVECs and iLECs treated with CB-839 and R162. n = 3 replicates. *P*-values were determined using a Welch's *t* test. **P*-value < 0.05. Data are represented as mean ± SD. **(B)** Migration rates of CB839- and R162-treated HUVECs and iLECs, normalized to the respective control. Error bars denote SD. n = 3 replicates. *P*-values were determined using a Welch's *t* test. **P*-value < 0.05. Data are represented as mean ± SD. **(C)** Migration rates of HUVECs and iLECs treated with R162 and dimethyl α-ketoglutarate (dmKG). **(D)** Representative pictures of vessel sprouting assay and total sprout length normalized to the respective control of HUVECs and iLECs treated with CB839, R162, DMaKG, and R162 + DMaKG. n = 3–8 replicates. *P*-values were determined using a Welch's *t* test. **P*-value < 0.05. Data are represented as mean ± SD. Scale bar = 30 *μ*m. Controls are the same as in (Fig 5).

migration only in the long-term (Zhan et al, 2016). And DMTU slightly increases the migration rate of iLECs significantly and HUVECs nonsignificantly when the cells are treated right after scratching (Fig 5G).

We further tested whether DMTU or NONOate supplementation could rescue sprouting defects caused by Rotenone. We found that neither DMTU nor NONOate can rescue sprouting defects in Rotenone-treated iLECs (Fig 5H). DMTU even further reduces total sprout length in Rotenone-treated HUVECs (Fig 5H). NONOate, however, increases the total sprout length of Rotenone-treated HUVECs by around twofold (Fig 5H). Taken together, migrating and sprouting HUVECs are more dependent than iLECs on adequate heme availability, likely for NO production. In contrast, iLECs are more sensitive to elevated ROS levels when migrating and Complex I activity appears to be crucial for sprouting beyond elevated ROS levels because of NONOate nor DMTU supplementation does not

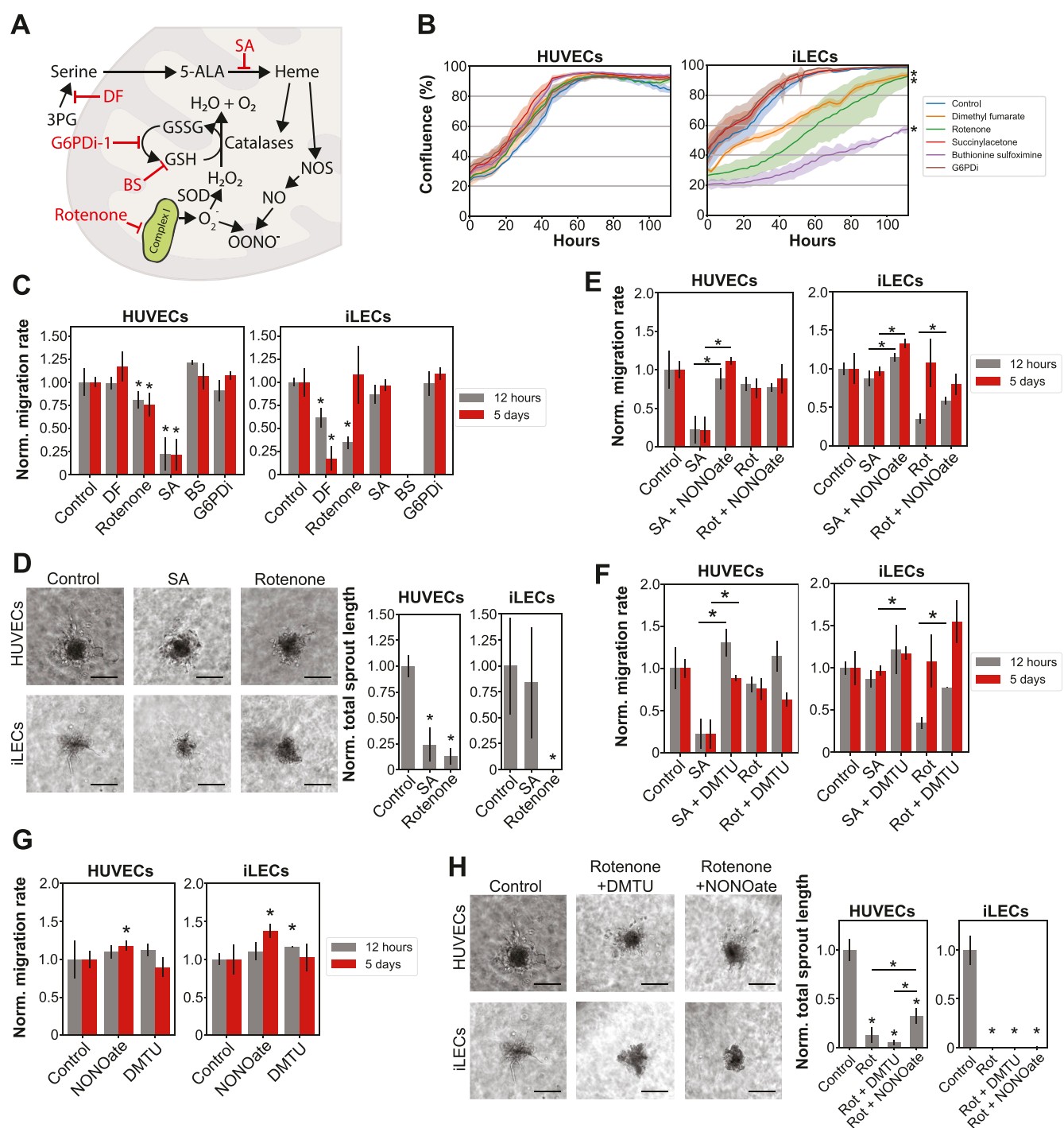

**Figure 5.  HUVECs and iLECs have divergent dependencies to nitric oxide and reactive oxygen species (ROS) metabolic pathways for migration and sprouting.**
**(A)** Schematic overview of the targets of the five drugs targeting ROS and NO metabolism. SA, succinyl acetone; DF, dimethyl fumarate; BS, buthionine sulfoxime. **(B)** Growth curves of HUVECs and iLECs treated with ROS and NO metabolism targeting drugs. n = 3 replicates. *P*-values were determined using a Welch's *t* test. *\*P*-value < 0.05. **(C)** Migration rates of HUVECs and iLECs treated with ROS and NO metabolism targeting drugs, normalized to the respective control. n = 3 replicates. *P*-values were determined using a Welch's *t* test. *\*P*-value < 0.05. No *P*-value calculation for BS-treated iLECs because BS-treated iLECs did not grow and scratch assay thus not possible. Data are represented as mean ± SD. **(D)** Representative pictures of vessel sprouting assay and total sprout length normalized to the respective control of HUVECs and iLECs treated with SA and rotenone. n = 3–8 replicates. *P*-values were determined using a Welch's *t* test. *\*P*-value < 0.05. Data are represented as mean ± SD. Scale bar = 30 *μ*m. Controls are the same as in Fig 4. **(E)** Migration rates of SA- and rotenone-treated HUVECs and iLECs supplemented with NONOate. **(F)** Migration rates of SA- and rotenone-treated HUVECs and iLECs supplemented with DMTU. **(G)** Migration rates of HUVECs and iLECs treated with NONOate or DMTU. **(H)** Representative pictures of vessel sprouting assay and total sprout length normalized to the respective control of HUVECs and iLECs treated with rotenone and supplemented with either NONOate or DMTU. n = 3–8 replicates. *P*-values were determined using a Welch's *t* test. *\*P*-value < 0.05. Data are represented as mean ± SD. Scale bar = 30 *μ*m. Controls are the same as in Fig 4.

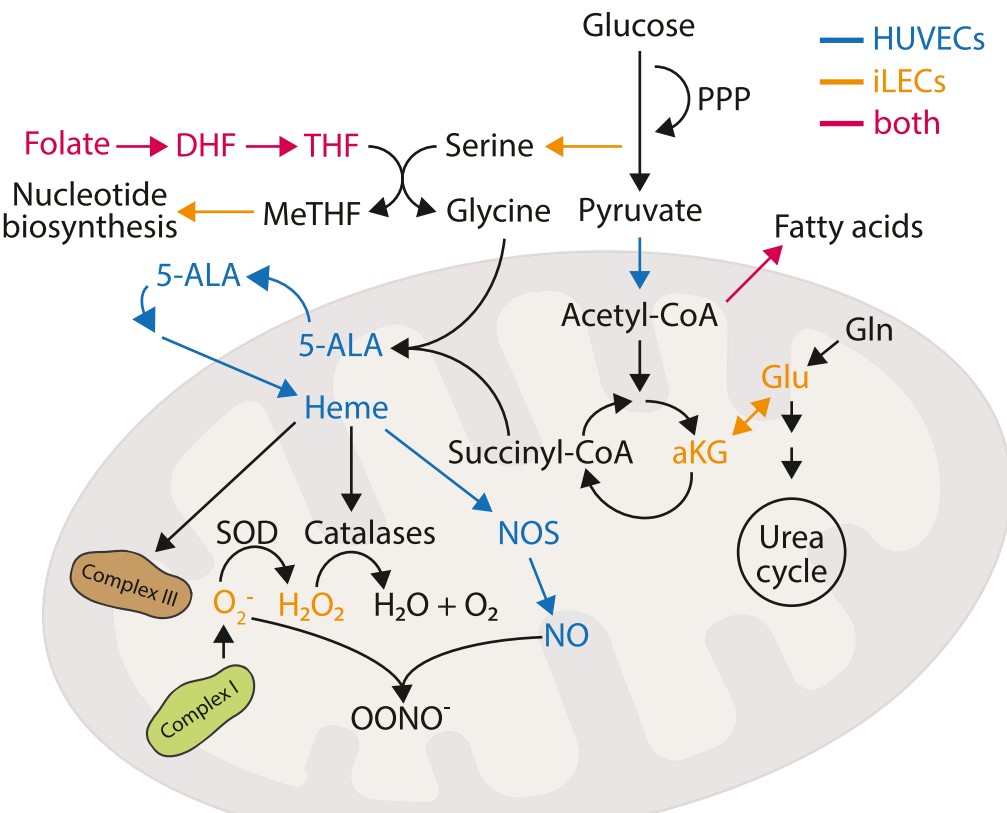

**Figure 6. Schematic overview of HUVEC- and iLEC-specific dependencies on metabolic pathways/reactions for migration.**

rescue Rotenone-induced sprouting defects in iLECs (Fig 5H). These results functionally confirm and back up the proteomics and metabolomics results, in which increased intermediates levels in heme biosynthesis only in HUVECs and a stronger enrichment of the ETC in iLECs proposed a cell type-specific dependence on these two pathways (Fig 6).

## Discussion

Employing a multi-omics approach, we elucidated the molecular patterns underlying the identities and quiescence induction dynamics of four different endothelial cell (EC) types. Our research aimed to create a comprehensive resource to enhance the understanding of EC biology. One of the key findings of our study is the demonstration that metabolic pathways are involved in the formation and maintenance of lymphatic endothelial cells (LECs) and blood endothelial cells (BECs) identities, states and functions (Fig 6). Whereas all cell lines exhibited similar growth and quiescence induction rates and phenotypes, our data revealed diverse underlying metabolic patterns. These patterns were reflected in cell type-specific uptake and secretion of metabolites and a significant intracellular metabolic and proteomic reorganization. Notably, all EC types secreted lactate, (iso) citrate and uracil and took up pyruvate, fumarate, and ascorbate during both proliferation and quiescence. However, the uptake of glucose and amino acids varied significantly between cell types and states, suggesting distinct metabolic patterns that support the specific functions of each EC type.

Moreover, intracellular metabolomics analysis identified a multitude of metabolites and metabolic pathways that changed between quiescence and proliferation in at least one cell line, highlighting EC-specific metabolic adaptations. These findings were corroborated by proteomics data, although changes in protein expression only partially accounted for variations in metabolite abundances, suggesting the involvement of other regulatory mechanisms, such as allosteric regulations or post-translational modifications.

Despite this, we identified a core metabolic signature at the protein level for quiescence in all EC types, characterized by the down-regulation of nucleotide metabolism and the upregulation of FAO and branched-chain amino acid (BCAA) catabolism. These results support the significant role of FAO in quiescence maintenance in HUVECs and propose FAO as a universal metabolic driver of EC quiescence maintenance (Kalucka et al, 2018). Furthermore, the increased expression of BCAA catabolism proteins suggests that BCAA catabolism plays a relevant role in EC quiescence maintenance, consistent with previous reports on hematopoietic stem cells and HUVECs (Liu et al, 2018; Andrade et al, 2021). Overall, our study strengthens the view that quiescent endothelial cells are metabolically active, consistent with similar findings in research on quiescent fibroblasts and epithelial cells (Coller et al, 2006; Coloff et al, 2016).

Our data provide further evidence that ECs from different tissues and vascular beds are metabolically distinct, and that metabolic rewiring during quiescence is driven by the specific needs of each EC type to support their unique functions. For instance, HUVECs and iLECs exhibited a range of phenotypic alterations in response to targeted inhibition of nucleotide and folate metabolism, fatty acid synthesis, TCA cycle, ROS and nitric oxide (NO) metabolism, and glutamate metabolism.

Fatty acid synthase (FASN) was previously reported to be crucial for the proliferation and vessel sprouting of HUVECs by preventing the accumulation of malonyl-CoA (Bruning et al, 2018). Our results indicate that FASN inhibition impairs proliferation and migration of both, HUVECs and iLECs, suggesting a conserved role of a fully functioning FASN in the angiogenesis of blood and lymphatic vessels. Moreover, inhibition of the mitochondrial pyruvate carrier (MPC) reduced proliferation of HUVECs but not iLECs, indicating that glycolysis-derived pyruvate, alongside acetate from FAO, serves as an additional source for fueling the TCA cycle and subsequent nucleotide precursor biosynthesis (Schoors et al, 2015). Decreased proliferation might also explain why short-term treatment with UK5099 reduced the migration of HUVECs but not iLECs. Interestingly, MPC inhibition over 5 d before the scratch assay impaired the migration of both HUVECs and iLECs, possibly by disrupting the TCA cycle and OXPHOS needed for the transition into and maintenance of quiescence (Petrelli et al, 2023).

Our inhibitor screen revealed that GLUD1 inhibition by R162 significantly reduces iLEC migration only after long-term treatment. Surprisingly, dmKG supplementation also reduces iLEC migration, potentially due to toxic intracellular glutamate accumulation (Miura et al, 1992; Parfenova et al, 2006). HUVECs, however, are resistant to glutamate accumulation, possibly due to increased glutamate usage or clearance mechanisms, as seen in brain endothelial cells and in vivo wall shear stress situations (Domoki et al, 2008; Hinca et al, 2021; Simões-Faria et al, 2025). Both R162 and dmKG impair sprouting in HUVECs and iLECs, suggesting a broader impact of glutamate metabolism on angiogenesis beyond migration. Previous reports propose that overactivation of ionotropic glutamate receptors, such as NMDA, results in decreased tube network formation and increased vascular permeability in brain BECs and HUVECs (Vazana et al, 2016; Sailem & Al Haj Zen, 2020). Increased glutamate levels are potentially cleared by increased secretion of glutamate in sprouting situations, which leads to higher auto- and paracrine activation of glutamate receptors, resulting in impaired sprout formation.

One of the most striking distinctions is HUVECs' reliance on heme biosynthesis, which is crucial for nitric oxide (NO) production and migration. Whereas NONOate supplementation promotes iLEC migration, indicating the general importance of NO, its role in vessel sprouting is complex (Murohara et al, 1999; Cooke & Losordo, 2002). NONOate partially rescues sprouting defects in Rotenone-treated HUVECs but does not affect migration, suggesting that sprouting involves more intricate mechanisms than migration, including extracellular cues and the mitigation of elevated superoxide levels (Wang et al, 2020).

Our findings contribute to the understanding of how metabolism governs the cellular states and functions of ECs from different tissues and vascular beds. We demonstrate that certain metabolic pathways distinctly influence the establishment and maintenance of HUVEC and iLEC phenotypes, potentially due to varying nutrient and oxygen availability in their microenvironments. Despite the comprehensive insights provided by this study, several limitations should be noted. Firstly, the in vitro nature of our experiments may not fully capture the complex in vivo microenvironment and its influence on EC metabolism. The metabolic interactions between ECs and other cell types within the tissue context were not addressed, potentially overlooking critical regulatory mechanisms. One important question is on the role of the microenvironment to modulate NO levels and influence vessel sprouting. Secondly, while we identified significant metabolic pathways involved in quiescence and proliferation, the causal relationships and underlying molecular mechanisms require further investigation. For instance, it remains unclear what mechanism enables HUVECs to clear glutamate more efficiently. In addition, our study focused on a limited number of EC types; thus, the findings may not be generalizable to all EC populations across different tissues and vascular beds. The long-term effects of metabolic inhibitors were not explored, leaving questions about potential adaptive responses or compensatory mechanisms in ECs. Finally, future studies should investigate novel metabolic interventions that could potentially modulate EC functions in disease contexts, as it has been done with LECs and tumor-associated ECs (García-Caballero et al, 2019; Zhang et al, 2023).

# Materials and Methods

### Cell culture

HUVECs were purchased from Lonza (cat. no. C2519A; mixed donor), HDBECs (cat. no. C-12211; single donor), and HDLECs (cat. no. C-12216; single donor) from PromoCell. Human intestinal samples were obtained from surplus discarded tissue of single donors from the University of Lausanne Hospital and are a kind gift from Tatiana Petrova from the University of Lausanne. All patients gave informed consent to have surgically resected tissue collected. The study was approved by the Canton of Vaud Ethics Commission for Research on Human Subjects and was conducted in accordance with the principles of the Declaration of Helsinki. All endothelial cells were cultured in T-75 cell culture flasks (cat. no. 156472; Thermo Fisher Scientific) in Endothelial Cell Growth Medium-2 (EGM-2) (BulletKit, cat. no. CC-3162; Lonza). EGM-2 was made from Endothelial Cell Basal Medium-2 (EBM-2), with added endothelial supplements including 2% FBS (vol/vol), hydrocortisone, VEGF, human FGF, R3-IGF-1, ascorbic acid, human EGF, glutaraldehyde GA-1000, and heparin. In addition, we added 10 mg/ml ciprofloxacin (cat. no. 17850; Sigma-Aldrich). After the first passage, regular FBS in the culture medium was replaced by dialyzed FBS (dFBS, cat. no. F0392; Sigma-Aldrich). Experiments were performed until passage six. For metabolomics, proteomics and phenotype experiments, cells were seeded at a density of 20,000 cells/cm$^2$ in a mixed medium (MM). MM consists of 50% EGM2 (with dFBS) and 50% medium 199 (cat. no. 22340020; Gibco) supplemented with 20% dFBS, 1% penicillin/streptomycin (P/S, cat. no. 15140122; Gibco),

2 mM glutamine (cat. no. 25030081; Gibco) and 0.4% (vol/vol) endothelial cell growth supplement (cat. no. C-39215; PromoCell).

## Quiescence induction and cell cycle analysis

ECs were grown for 10 d to assess fractions of proliferating and quiescent cells at each day. The fraction was determined by EdU incorporation into DNA using the EdU Flow Cytometry Kit 488 from baseclick (Salic & Mitchison, 2008) (cat. no. BCK-FC488-100; Sigma-Aldrich). Briefly, EdU was added at a concentration of 10 $\mu$M 24 h prior to trypsinization and cell fixation with 4% PFA. Using a click-it reaction, 6-FAM was attached to EdU and EdU incorporation analyzed using a BD LSRFortessa Cell Analyzer with a 488 nm laser for excitation and a 530/30 emission filter. Flow cytometry data were analyzed with Flowing Software 2.5.1 from Turku Bioscience. Viability of cells was determined by Trypan blue staining and measurement on a Countess II Automated Cell counter (Thermo Fisher Scientific). Functional quiescence was determined by growing the cells for 10 d, seeding them at 20,000 cells/cm$^2$ in mixed medium and measurement of growth using a Incucyte S3 Live-Cell Analysis platform (Sartorius).

## Proteomics

Proteomics sample preparation protocol was adapted from Villén and Gygi (2008) and Weekes et al (2014). In brief, cells were grown in 1.5 ml mixed medium in six-well plates as described above. After removal of medium and washing of cells with PBS, cells were lysed in lysis buffer (8 M urea buffer) and subsequently sonicated, centrifuged and supernatants transferred into fresh tubes for protein reduction with DTT and alkylation with iodoacetamide. Supernatants were incubated for 25 min at 50°C with 5 mM DTT and after cooling down to room temperature, iodoacetamide to 15 mM final concentration was added and the mixture incubated for 30 min at RT in the dark. An additional 5 mM DTT was used to quench unreacted iodoacetamide for 15 min at RT in the dark. Protein concentration was determined by BCA assay. Samples were then diluted 1:8 with 100 mM HEPES, pH 8.5, to reduce the concentration of urea to 1 M. Trypsin was added to the diluted samples to reach a 100:1 sample:trypsin (w/w) ratio and the mix incubated at 37°C overnight on a thermomixer. To stop trypsin digestion, samples were acidified with TFA to 0.4% (vol/vol) to reach a pH around 3. Before MS measurements, samples were desalted using Pierce C18 spin columns (cat. no. 89870; Thermo Fisher Scientific).

Peptides were analyzed online by liquid chromatography-tandem mass spectrometry (LC-MS/MS). Online reversed phase chromatography was performed using a Vanquish Neo UPLC system (Thermo Fisher Scientific) equipped with a heated column compartment set to 50°C. Mobile Phase A consisted of 0.1% formic acid (FA) in water, whereas Mobile Phase B was 80% acetonitrile in water and 0.1% FA. Peptides (~1 $\mu$g) were loaded onto a C18 analytical column (500 mm, 75 $\mu$m inner diameter), packed in-house with 1.8 $\mu$m ReproSil-Pur C18 beads (Dr. Maisch) fritted with Kasil, keeping constant pressure of 600 bar or a maximum flow rate of 1 $\mu$l/min. After sample loading, the chromatographic gradient was run at 0.3 $\mu$l/min and consisted of a ramp from 0% to 43% Mobile Phase B in 70 min, followed by a wash at 100% Solution B in 9 min total, and a final re-equilibration step of 3 column volumes (total run time 90 min).

Peptides from each sample were analyzed on an Orbitrap HF-X mass spectrometer (Thermo Fisher Scientific) using an overlapping window data-independent analysis (DIA) pattern described by Searle et al (2018), consisting of a precursor scan followed by DIA windows. Briefly, precursor scans were recorded over a 390–1,010 m/z window, using a resolution setting of 120,000, an automatic gain control (AGC) target of $1 \times 10^6$ and a maximum injection time of 60 ms. The RF of the ion funnel was set at 40% of maximum. A total of 150 DIA windows were quadrupole selected with an 8 m/z isolation window from 400.43 to 1,000.7 m/z and fragmented by higher energy collisional dissociation, HCD, (NCE = 30, AGC target of $1 \times 10^6$, maximum injection time 60 ms), with data recorded in centroid mode. Data were collected using a resolution setting of 15,000, a loop count of 75 and a default precursor charge state of +3. Peptides were introduced into the mass spectrometer through a 10 $\mu$m tapered pulled tip emitter (Fossil Ion Tech) via a custom nano-electrospray ionization source, supplied with a spray voltage of 1.6 kV. The instrument transfer capillary temperature was held at 275°C.

All Thermo RAW files were converted to mzML format using MSconvert from the ProteoWizard package (Chambers et al, 2012) (version 3.0.2315). Vendor-specific peak picking was selected as the first filter and demultiplexing with a 10 ppm window was used for handling the overlapping data collection window scheme. Processed mzML files were then searched using DIA-NN (version 1.8) (Demichev et al, 2020) and the UniProt *Homo sapiens* proteome (UP000005640, 15 June 2021) as the FASTA file for a "library-free" deep neural network-based search approach. Data were searched using deep learning-based spectra and retention time as described by Demichev et al (2020), with trypsin as the protease, and allowing for two missed cleavages, with N-terminal methionine cleavage, and cysteine carbamidomethylation. Peptide length was allowed to range from 7–30 amino acids with a precursor charge state range from +1 to +4, a precursor range of 300–1800 m/z and a fragment ion range of 200–1800 m/z. Data were processed to a 1% precursor-level FDR with mass accuracy, MS1 accuracy, and match between runs set to the software default settings. A single-pass mode neural network classifier was used with protein groups inferred from the input *H. sapiens* FASTA file. Protein quantities were normalized by delayed normalization and maximal peptide ratio extraction (maxLFQ) (Cox et al, 2014). All downstream data analysis was performed using Python (Version 3.8 or higher). Differential analysis was performed using a *t* test and resulting *P*-values were corrected for FDR by the Storey and Tibshirani method (Storey & Tibshirani, 2003), and a *Q*-value < 0.05 and an absolute ($\log_2$(FC)) > 0.5 was considered significant. Pathway enrichment analysis was performed using Reactome (Fabregat et al, 2018).

## Intracellular metabolomics

Cells were grown in 1.5 ml mixed medium in six-well plates as described above. Every 24 h, the medium was removed from the

wells and cells were washed with pre-warmed wash buffer, made of freshly prepared 75 mM ammonium carbonate in nanopure water, adjusted to pH 7.4 using 10% acetic acid and pre-heated to 37°C. After washing the cells, metabolites were extracted with ice-cold extraction buffer, containing 40% (vol/vol) methanol, 40% (vol/vol) acetonitrile, and 20% (vol/vol) nanopure water for 1 h at –20°C. Cells were detached from the wells using a cell lifter, transferred into tubes and centrifuged. Supernatants (metabolic extracts) were stored at –80°C until measurement. Untargeted metabolomics of metabolic extracts was performed by flow injection analysis–time-of-flight mass spectrometry on an Agilent 6550 Q-TOF mass spectrometer as previously described (Fuhrer et al, 2011). The mobile phase was 60:40 isopropanol/water supplemented with 1 mM $NH_4F$ at pH 9.0, as well as 10 nM hexakis (1H, 1H, 3H-tetrafluoropropoxy) phosphazine and 500 nM 3-amino-1-propanesulfonic acid for online mass calibration. The injection sequence was randomized. Measurements were performed in negative ionization mode, and spectra were recorded from a mass/charge ratio of 50–1,000 at 1.4 Hz. Ions were annotated based on their measured mass using reference compounds from the Human Metabolome Database (HMDB 4.0), with a tolerance of 1 mDa. Data analysis was performed with an in-house developed pipeline based on Matlab (Version R2021B; The MathWorks). Samples were normalized with a moving median-based temporal drift normalization, followed by a normalization of the mean ion intensity within each cell type to account for the cell number differences at sampling. Differential analysis was performed using a $t$ test and significance was corrected for multiple hypothesis testing with the Benjamini-Hochberg method (Benjamini & Hochberg, 1995), and an adjusted $P$-value < 0.05 and an absolute ($\log_2$(FC)) > 0.5 was considered significant. Metabolic pathway enrichment was done using pathway definitions from The Small Molecule Pathway Database (SMPDB), with a $P$-value cutoff of 0.05 and an absolute ($\log_2$(FC)) cutoff of 0.25. Significance of enrichments was corrected for multiple hypothesis testing by the Benjamini-Hochberg method, and an adjusted $P$-value of < 0.05 was considered significant.

## Extracellular metabolomics

Cells were grown in 1.5 ml mixed medium in six-well plates as described above. Every 24 h, the medium was replaced with fresh medium. Supernatant samples were taken 0, 2, 22, and 24 h after each medium exchange. Supernatant samples were diluted 1:50 with nanopure water before metabolomics measurement. Untargeted metabolomics of supernatant samples was performed by flow injection analysis–time-of-flight mass spectrometry on an Agilent 6520 Q-TOF mass spectrometer as previously described (Fuhrer et al, 2011). Ions were annotated based on their measured mass using reference compounds from the Human Metabolome Database (HMDB 4.0), with a tolerance of 3 mDa. All downstream data analysis was performed using Python (Version 3.8 or higher). Within each day, samples were normalized to the first time point and linear regression applied to determine the uptake or secretion rate as normalized ion intensity per hour.

## Functional assays

For conditions with pharmacological perturbations, we used IC50 concentrations of inhibitors previously noted for HUVECs or, if missing for endothelial cells, other mammalian cell lines (Table S5) (Heller et al, 2001; Costantini et al, 2010; Mader et al, 2012; Gross et al, 2014; Focaccetti et al, 2015; Jin et al, 2015; Rae et al, 2015; Nishizawa et al, 2018; Wang et al, 2018; Liu et al, 2019; Ghergurovich et al, 2020; Ocaña et al, 2023). The drugs or supplements were added to mixed medium at the indicated concentration, and the medium was sterile filtered through a filter with 0.2 $\mu$m pore size. Medium with drugs was refreshed after 2 d and prepared freshly for each experiment. For drugs dissolved in DMSO, a control condition medium with equal concentration of DMSO was prepared.

## Phenotypic characterizations

Phenotype was assessed using three characteristics: response of cells to drug treatment in proliferation, migration, and sprouting.

### Proliferation
To determine the effect of drug compounds on EC proliferation, iLECs and HUVECs were seeded in 96-well plates at 20,000 cells/$cm^2$ in mixed medium on day zero. Cells were left to attach for four to 6 h, after which the medium was exchanged with drug-containing medium. Cell growth was determined using an Incucyte S3 Live-Cell Analysis instrument (Sartorius). Statistical significance was determined by normalizing all time points of drug-treated samples to the non-treated control samples, subsequent calculation of areas under the curve in the biological triplicates and applying a Welch's $t$ test.

### Migration
iLECs and HUVECs were seeded in 96-well plates at 20,000 cells/$cm^2$ and grown for 5 d with or without drugs, to reach contact-inhibited quiescence. On day five, a scratch wound was inflicted on the confluent cell layer in each well using the tip of a 20 $\mu$l pipette. Medium with drugs was then immediately added to cells which had been grown to contact inhibition with drugs, respectively, newly added to cells which had been grown to quiescence without drugs to capture the effects of chronic and acute drug exposure on migration. The scratch in each well was imaged for 48 h using an Incucyte S3 Live-Cell Analysis instrument (Sartorius). Migration rate was determined by measuring the width of the scratch at 0 and 12 h using Fiji ImageJ (version 1.54d) and calculating the distance that was closed by cells in 12 h (Schindelin et al, 2012).

### Vessel sprouting
Vessel sprouting assays were performed as previously described (Tetzlaff & Fischer, 2018). Methocel solution was prepared by dissolving 1.2% methyl cellulose (cat. no. M0512; Sigma-Aldrich) in basal medium 199 and stirring the solution at 4°C overnight, followed by centrifugation at 3,500$g$ for 3 h at 4°C. Collagen solution was prepared at the time of the experiment from 3.75 mg/ml PureCol collagen (cat. no. 5006; Sigma-Aldrich) in 0.1% acetic acid. The collagen solution was mixed 8:1 with mixed medium, after which pH was adjusted with 0.2 M NaOH and 1 M HEPES. Spheroids

were generated according to the same published protocol by suspending HUVECs or iLECs in mixed medium with 20% Methocel solution to 16,000 cells/ml. From the cell suspension, droplets of 25 µl were pipetted onto the lid of a petri dish. The droplets were incubated hanging upside-down for 24 h at 37°C to allow for the formation of spheroids. After 24 h, spheroids were collected by gently washing them off the petri dish lid with medium. Spheroids were then centrifuged for 5 min at 100*g* and carefully resuspended in Methocel solution with 20% FBS and 0.2% Penicillin/ Streptomycin. The cell suspension was then gently mixed 1: 1 with the collagen solution and distributed over wells of a 48-well plate. The plate was incubated for 30 min to allow gelation, after which spheroids were stimulated with mixed medium containing 350 ng/ml human VEGF-165 recombinant protein (cat. no. PHC9391; Thermo Fisher Scientific), for a final concentration of 50 ng/ml in the gel. Depending on the condition, drug compounds were added to the medium at 7× of the desired final concentration in the gel. Spheroids gels were imaged by brightfield microscopy after 72 h on an Invitrogen EVOS M5000 Imaging System at 40× magnification and RT as previously described (Heiss et al, 2015). Sprouts were defined by visual inspection and measured in length using Fiji ImageJ (Schindelin et al, 2012).

## Data Availability

Raw mass spectrometry data have been deposited in the MassIVE database (https://massive.ucsd.edu/ProteoSAFe/static/ massive.jsp) under the accession number MSV000096399 for the proteomics data and MSV000096398 for the metabolomics data. Spheroid pictures are available on BioStudies (https:// www.ebi.ac.uk/biostudies/) with the accession number S-BSST1716.

## Supplementary Information

## Acknowledgements

We thank the Flow Cytometry Core Facility of ETH Zurich for support with flow cytometric analyses. This work was supported by the Swiss National Science Foundation (CRSII5_177191 to N Zamboni).

### Author Contributions

S Durot: conceptualization, formal analysis, validation, investigation, visualization, methodology, and writing – original draft, review, and editing.
PF Doubleday: data curation, methodology, and writing – original draft.
L Schulla: validation.
A Sabine: methodology.
TV Petrova: resources and supervision.
N Zamboni: conceptualization, resources, supervision, funding acquisition, project administration, and writing – original draft, review, and editing.

### Conflict of Interest Statement

The authors declare that they have no conflict of interest.

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
