## [Reviewer comments · Life Science Alliance]

Multi-omics analysis reveals metabolic diversity underlying endothelial cell functions

Stephan Durot, Peter Doubleday, Lydia Schulla, Amelie Sabine, Tatiana Petrova, and Nicola Zamboni
DOI: <https://doi.org/10.26508/lsa.202503526>

Corresponding author(s): Nicola Zamboni, ETH Zurich

Review Timeline:

Submission Date:	2025-10-09
Editorial Decision:	2025-11-13
Revision Received:	2025-12-04
Accepted:	2025-12-08

Scientific Editor: Sarita Hebbar

Transaction Report:

Please note that the manuscript was reviewed at *Review Commons* and these reports were taken into account in the decision-making process at *Life Science Alliance*.

Review
COMMONS

Review #1

1. Evidence, reproducibility and clarity:

Summary:

The study by Durot and colleagues explores the metabolic heterogeneity of endothelial cells (ECs) across distinct subtypes (blood vs. lymphatic) and growth states (proliferating vs. quiescent). Through integrated proteomic and metabolomic analyses, the authors demonstrate that quiescent ECs are not metabolically inactive but instead undergo subtype-specific metabolic reprogramming. Functional perturbation of key metabolic pathways using chemical inhibitors results in differential phenotypic responses in blood versus lymphatic ECs. Collectively, the findings underscore a critical, context-dependent role of metabolism in maintaining EC function and highlight metabolic specialization as a fundamental feature of endothelial diversity.

General Comments:

This manuscript presents a comprehensive and methodologically robust investigation into the metabolic diversity of cultured ECs. By combining proteomic and metabolomic approaches, the authors provide novel insights into the distinct metabolic profiles of blood and lymphatic ECs, and how these profiles shift as ECs transition from a proliferative to a quiescent state. The observation that quiescent ECs exhibit active metabolic reprogramming, rather than simply entering a dormant state, is particularly compelling and challenges existing models of cellular quiescence.

The work is timely, well-written and addresses a significant gap in our understanding of endothelial metabolism. The integration of large-scale omics data with functional perturbation experiments strengthens the overall conclusions and enhances the impact of the study.

Nevertheless, while the data are largely convincing, certain experimental aspects-particularly those related to the in vitro sprouting assays-require further validation to solidify the mechanistic interpretations. Additionally, some findings would benefit from further validation using alternative approaches (e.g., chemical perturbation studies).

Specific Comments:

1. Image quality in sprouting assays - The images presented for the sprouting assays (e.g., Figure 4) are of suboptimal resolution and quality, making it difficult to evaluate the effects of the various compounds on EC behavior. Even under control conditions, clear sprout-like structures are not readily discernible. Improved image resolution-preferably through high-quality bright-field microscopy-and the inclusion of immunofluorescence images of labeled endothelial spheroids are recommended to enhance interpretability.
2. Validation of the quiescence model - The current approach to induce quiescence should be further substantiated. Beyond proliferation markers, additional hallmarks of quiescent cells-such as epigenetic signatures, protein quality control mechanisms, and translational activity-should be assessed to confirm that the EC subtypes achieve a bona fide resting state.
3. Reversibility of quiescence - It is important to demonstrate that the EC subtypes investigated can re-enter the cell cycle following release from contact inhibition. Without such evidence, the possibility remains that some of the observed metabolic features reflect a transition to senescence rather than reversible quiescence.
4. Assessment of cell viability - While EC proliferation, migration, and sprouting were examined to infer functional roles of metabolic adaptations, analyses of cell viability and death are also necessary to evaluate potential homeostatic or survival-related functions of the observed metabolic changes.
5. Validation of pharmacological findings - The pharmacological inhibition experiments are informative and constitute a central part of the study. However, given the possibility of off-target effects, key conclusions should be corroborated using alternative loss-of-function approaches, such as RNA interference (e.g., shRNA or siRNA).

2. Significance:

In summary, this manuscript makes a substantial contribution to the field and is likely to stimulate further research into endothelial metabolic regulation. With additional experimental validation, the study has the potential to serve as a reference in both vascular and metabolic research.

Review #2

1. Evidence, reproducibility and clarity:

By employing a proteomics and metabolomics approach the authors clarified the molecular landscape of 4 major EC types in quiescent and proliferating conditions. The study is extensive and adds novelty to the EC research

Major comments:

- it was not clear whether the authors worked with single donor endothelial cells or with mixed donors. This should be clarified as it is important for the statistical analyses (single donor based EC research typically uses n=4, while for the mixed donor, an n=3 is sufficient).
- I would like to see a sentence on the importance of shear stress in EC behavior (metabolism) in the introduction. It was recently shown that the in vivo situation of ECs encountering wall shear stress (Faria et al, PMID: 39832080) affects the metabolic behavior switching to glutamine metabolism. This aligns with the research of the authors as well.
- suggestion for the authors: it could be useful if a figure is introduced to show the "physiological" location of the 4 EC used and that a rationale is provided for this.
- figures are of low quality, I found it very difficult to see the spheroid/sprouting images. This should be addressed in the final version prior publication.
- Fig 2 c: I'm not sure if this panel is very relevant, when looking into detail, opposite pathways are present (glycolysis - gluconeogenesis). As well, I'm not sure if galactose metabolism is truly relevant, unless the author managed to measure distinct hexose and hexose-phosphates? Given the flow injection analysis setup, I doubt this. Would suggest to move this to supplement or to simply leave it out.
- Fig 3 b: was there any statistics performed on these data to compare the different setups?

2. Significance:

the study adds insights to the ongoing research on EC molecular behavior.

using different types of ECs in both quiescent and proliferating mode, as well as the validation of pathways by introducing inhibitors combined with the sprouting assays is an asset.

I would like to see stated the biological complexity of EC, it was recently shown that shear stress plays an important role in EC metabolism.

Corresponding author(s): Nicola Zamboni

1. General Statements [optional]

We thank the reviewers for their kind and constructive comments. We are happy to read that the reviewers found our study methodologically robust and comprehensive in addressing the metabolic heterogeneity of endothelial cells.

Reviewer 1, comment 1: Image quality in sprouting assays - The images presented for the sprouting assays (e.g., Figure 4) are of suboptimal resolution and quality, making it difficult to evaluate the effects of the various compounds on EC behavior. Even under control conditions, clear sprout-like structures are not readily discernible. Improved image resolution-preferably through high-quality bright-field microscopy-and the inclusion of immunofluorescence images of labeled endothelial spheroids are recommended to enhance interpretability.

Response:

We appreciate the reviewer's concern and have revisited the sprouting assay images. Our approach is consistent with established methods in the field (Heiss et al., FASEB J, 2015), where brightfield imaging is routinely used for quantification without additional immunostaining. Hence, we believe that the brightfield images are of sufficient resolution to allow reproducible quantification of normalized total sprout length. All experiments were performed under identical imaging and analysis protocols, and thus we are confident that the quantification reflects true biological differences. We cite the reference in the revised manuscript and clarify it as well in the Methods section.

Reviewer 1, comment 2: Validation of the quiescence model - The current approach to induce quiescence should be further substantiated. Beyond proliferation markers, additional hallmarks of quiescent cells-such as epigenetic signatures, protein quality control mechanisms, and translational activity-should be assessed to confirm that the EC subtypes achieve a bona fide resting state.

Response:

We acknowledge the value of proper phenotyping of quiescent cells. However, most studies involving quiescent (endothelial) cells rely on EdU incorporation or similar proliferation markers to confirm entry into a non-proliferative state (Kalucka et al., Cell Metabolism, 2018; Coloff et al., Cell Metabolism, 2016). In our study, we have used EdU staining and FACS analysis to establish cell cycle arrest. Moreover, we find clear proteomic patterns that support the case of a quiescent state. We have also demonstrated the reversibility of quiescence (see Suppl. Fig. 1c) via reseeding and proliferation recovery of all EC types, which is a defining functional hallmark of true quiescence. Together, the EdU, proteomic and reseeding/proliferation data provide strong evidence that our EC subtypes reach a physiologically quiescent, non-senescent state.

Reviewer 1, comment 3: Reversibility of quiescence - It is important to demonstrate that the EC subtypes investigated can re-enter the cell cycle following release from contact inhibition. Without such evidence, the

possibility remains that some of the observed metabolic features reflect a transition to senescence rather than reversible quiescence.

Response:

This is an excellent suggestion. We have included new data that shows that ECs regain proliferative capacity upon reseeded of quiescent ECs at lower confluency (Suppl. Fig. 1c). The results support the interpretation that the observed metabolic features reflect reversible quiescence rather than senescence.

Reviewer 1, comment 4: Assessment of cell viability - While EC proliferation, migration, and sprouting were examined to infer functional roles of metabolic adaptations, analyses of cell viability and death are also necessary to evaluate potential homeostatic or survival-related functions of the observed metabolic changes.

Response:

We appreciate the Reviewer's concern about cell viability in our experimental setup, and we agree that viability assessment is important. Using trypan blue staining and automated cell counting, we observed that >85% of ECs remained viable from day 1 through day 10 of the quiescence model and included these results in the manuscript (Suppl. Fig. 1b).

Reviewer 1, comment 5: Validation of pharmacological findings - The pharmacological inhibition experiments are informative and constitute a central part of the study. However, given the possibility of off-target effects, key conclusions should be corroborated using alternative loss-of-function approaches, such as RNA interference (e.g., shRNA or siRNA).

Response:

We recognize the possibility of side effects for pharmacological inhibitors, but the inhibitors, including the ones that show the strongest different effects in HUVECs and iLECs (succinyl acetone and R162) in our study are well-established, selective inhibitors of glutamate dehydrogenase (Wang et al., *Pharmacological Research*, 2022) and δ -aminolevulinic acid dehydratase (Nauli et al., *J Clin. Biochem. Nutr.*, 2023), respectively, and have not been reported to exhibit significant off-target activity in endothelial cells. Furthermore, the aim of our study was not to define specific mechanistic pathways, but to highlight phenotype-specific metabolic vulnerabilities in distinct endothelial states. Performing knockdown experiments would go beyond the scope and focus of this manuscript and introduce their own limitations, including off-target effects and, most importantly, timing mismatches relative to our long-term assays (e.g., sprouting assays assessed at day 3 versus transient RNAi effects lasting for only 1-2 days). We hope the Reviewer agrees that our current approach sufficiently supports the study's conclusions.

Reviewer 2, comment 1: it was not clear whether the authors worked with single donor endothelial cells or with mixed donors. This should be clarified as it is important for the statistical analyses (single donor based EC research typically uses n=4, while for the mixed donor, an n=3 is sufficient).

Response:

We thank Reviewer 2 for highlighting that we did not include this information in the Methods section and we did so in the revised manuscript. HDBECs, HDLECs and iLECs are from single donors, HUVECs are from mixed donors. We acknowledge the reviewer's concern about the power of statistical analyses, but we think that n=3 is sufficient with proper correction for statistical tests. Furthermore, previous *in vitro* studies with ECs are done with single donor cells and in biological triplicates (Wong et al., 2017; Kalucka et al., 2018; Simões-Faria et al., 2025 and more). Moreover, for sprouting assays, we have n > 3 for most conditions.

Reviewer 2, comment 2: I would like to see a sentence on the importance of shear stress in EC behavior (metabolism) in the introduction. It was recently shown that the in vivo situation of ECs encountering wall shear stress (Faria et al, PMID: 39832080) affects the metabolic behavior switching to glutamine metabolism. This aligns with the research of the authors as well.

Response:

We thank Reviewer 2 for drawing our attention to this relevant and interesting study. We mention the study in the introduction and the discussion.

Reviewer 2, comment 3: suggestion for the authors: it could be useful if a figure is introduced to show the "physiological" location of the 4 EC used and that a rationale is provided for this.

Response:

We have included this in Supplementary Figure 1 and in the text.

Reviewer 2, comment 4: figures are of low quality, I found it very difficult to see the spheroid/sprouting images. This should be addressed in the final version prior publication.

Response:

The new version has higher quality sprouting images in figure 4 and 5. The images can also be found in high quality on BioStudies (Accession: S-BSST1716).

Reviewer 2, comment 5: Fig 2 c: I'm not sure if this panel is very relevant, when looking into detail, opposite pathways are present (glycolysis - gluconeogenesis). As well, I'm not sure if galactose metabolism is truly relevant, unless the author managed to measure distinct hexose and hexose-phosphates? Given the flow injection analysis setup, I doubt this. Would suggest to move this to supplement or to simply leave it out.

Response:

The reviewer is correct; the employed analytics cannot distinguish different hexoses and hexose-phosphates. We have moved figure 2c to supplementary figure 4c.

Reviewer 2, comment 6: Fig 3 b: was there any statistics performed on these data to compare the different setups?

Response:

We performed statistical analyses on this data and included it in the figures and figure legends.

November 13, 2025

RE: Life Science Alliance Manuscript #LSA-2025-03526-T

Nicola Zamboni
Institute of Molecular Systems Biology
ETH Zurich
Wolfgang-Pauli Str. 16
Zurich 8093
Switzerland

Dear Dr. Zamboni,

Thank you for submitting your revised manuscript entitled "Multi-omics analysis of endothelial cells reveals the metabolic diversity that underlies endothelial cell functions". Your revised manuscript was reviewed by the original reviewers whose comments are appended below. They find that your revised version is improved and has addressed their concerns satisfactorily. In line with the reviewers' evaluation, we would be happy to publish your paper in Life Science Alliance pending final revisions necessary to meet our formatting guidelines. Reviewer 2 has suggested some minor edits that we encourage you to pay attention to.

- Kindly confirm if images in Fig 4d (control; HUVECs and iLECs), Fig 5d (control; HUVECs and iLECs) and Fig 5h (control; HUVECs and iLECs) are different images. If the same image has been utilised, then please indicate as such in the figure legends of all the panels. In this case please also clearly state in the results section if these assays were run simultaneously and if they use the same control.
- For the human intestinal lymphatic endothelial cells (from the University of Lausanne), please provide details as suggested in our guidelines (<https://www.life-science-alliance.org/editorial-policies#humans>).
- Please provide details for imaging (name of microscope, and details for objective (NA, type), temperature of imaging).
- Please include a scale bar for image panels in Figures 4 and 5.
- Please upload all figure files as individual ones, including the supplementary figure files; all figure legends should only appear in the main manuscript file.
- Please add a Running Title and a Summary Blurb/Alternate Abstract in our system.
- You have the option of using Figure 6 as a Graphical Abstract in which case you would have to remove this figure file and upload it instead as a graphical abstract file, and edit the text accordingly.
- Please add ORCID ID for the corresponding author - you should have received instructions on how to do so
- Please add a Category for your manuscript in our system.
- Please add the X and Bluesky handles of your host institute/organization, as well as your own and/or one of the authors, in our system.
- Please add Author Contributions to our system as well.
- Please add your main, supplementary figure, and table legends to the main manuscript text after the references section.
- Please use the [10 author names, et al.] format in your references (i.e., limit the author names to the first 10).
- Please be sure that the authorship listing and order is correct.

A. FINAL FILES:

B. MANUSCRIPT ORGANIZATION AND FORMATTING:

Thank you for your attention to these final processing requirements. Please revise and format the manuscript and upload materials as soon as you are able.

Sincerely,

Sarita Hebbar, PhD
Scientific Editor
Life Science Alliance
<http://www.lsajournal.org>

Reviewer #1 (Comments to the Authors (Required)):

The revised manuscript is improved and I consider it acceptable for publication now. Overall, the work provides relevant data about the metabolic traits and behaviors of different types of vascular cells and will be of interest for the community.

Reviewer #2 (Comments to the Authors (Required)):

I believe the authors have addressed appropriately my comments.

I still have some minor comments:

- size bar should be added to all spheroid pictures (Fig4-d, Fig5-d,h).
- in Fig4 d it states dmaKG, while the corresponding bar graphs state dmKG. I would make this consistent.

December 8, 2025

RE: Life Science Alliance Manuscript #LSA-2025-03526-TR

Prof. Nicola Zamboni
ETH Zurich
Institute of Molecular Systems Biology, ETH Zurich
Otto-Stern-Weg 3
Zurich, Zurich 8093
Switzerland

Dear Dr. Zamboni,

Thank you for submitting your Research Article entitled "Multi-omics analysis reveals metabolic diversity underlying endothelial cell functions". It is a pleasure to let you know that your manuscript is now accepted for publication in Life Science Alliance. Congratulations on this interesting work.

Your manuscript will now progress through copyediting and proofing. At the proofs stage, we urge you to mention in Figure legend for panel 4D that the controls are shared with Figure 5D and 5H, as you have done for figure legend 5. Further we also encourage you to add details in the methods section for the imaging of spheroids as requested previously. It is journal policy that authors provide original data upon request.

DISTRIBUTION OF MATERIALS:

Again, congratulations on a very nice paper. I hope you found the review process to be constructive and are pleased with how the manuscript was handled editorially. We look forward to future exciting submissions from your lab.

Sincerely,

Sarita Hebbar, PhD
Scientific Editor
Life Science Alliance
<http://www.lsajournal.org>